# Optimizing of Traffic-Signal Timing Based on the FCIC-PI—A Surrogate Measure for Fuel Consumption

Suhaib Alshayeb [1,*], Aleksandar Stevanovic [2], Jelka Stevanovic [3] and Nemanja Dobrota [4]

1   CHA Consulting, 8935 NW 35th Ln, Doral, FL 33172, USA
2   Department of Civil & Environmental Engineering, Swanson School of Engineering, University of Pittsburgh, 341A Benedum Hall, 3700 O'Hara Street Pittsburgh, Pittsburgh, PA 15261, USA
3   Principal—Retime LLC, 225 Valley Park Dr, Pittsburgh, PA 15216, USA
4   Kittelson & Associates, Inc., 100 M St SE #910, Washington, DC 20003, USA
*   Correspondence: suhaibalshayeb2019@gmail.com

**Abstract:** Optimizing signal timing improves sustainability metrics (e.g., fuel consumption or "FC"). Historically, traffic agencies have retimed signal timing to improve mobility measures (e.g., delays). However, optimizing signals to reduce delays does not necessarily mitigate sustainability measures. Hence, this study introduces an approach that integrates a newly derived surrogate measure for FC, traffic microsimulation software, and a stochastic genetic algorithm. This approach optimizes signal timing to reduce the surrogate measure and reduce sustainability metrics. This study also evaluated the impact of heavy vehicles' presence in a fleet on signal timing and FC savings. A 13-intersection arterial on Washington Street in the Chicago metro area served as a case study. Optimized signal timing delivered solutions that balanced both sustainability and mobility. The estimated excess FC savings ranged between 8 and 12% under moderate operating conditions, with no heavy vehicles, compared to the initial signal timing. The savings reached up to ~14% when many heavy vehicles existed on the side streets. Most of the improvements came without worsening traffic-mobility efficiency, which shows the possibility of a fair tradeoff between mobility and sustainability. All optimization scenarios showed that a slightly longer cycle length than the one implemented in the field is required to reduce FC.

**Keywords:** signal-timing optimization; fuel consumption; emissions; surrogate measure; Performance Index

## 1. Introduction

Fuel consumption (FC) negatively impacts the environment, health, and the economy [1] by inducing global climate change, reducing the quality of life due to causing health issues (e.g., respiratory, nervous, and cardiovascular diseases), and causing inflation due to increasing oil prices, respectively [1]. The traffic congestion problem in metropolitan areas and rapid urbanization of rural regions are leading causes of excess fossil-fuel combustion [2].

Signalized intersections are one of the primary sources of excess vehicular FC [3]. They involve enormous traffic volumes, deceleration–acceleration events, and long idling times to accommodate the conflicts caused by multiple intersecting traffic streams. Moreover, unsustainable operational conditions (e.g., a large percentage of heavy vehicles and high cruising speeds) increase extra FC at those intersections [4]. Hence, it is crucial to use the most efficient countermeasures to mitigate traffic congestion by seeking more-sustainable transportation systems.

Traffic signals are fundamental in reducing congestion and easing moving through intersections [5]. Traditionally, researchers and practitioners have put much effort into developing signal timing that reduces motorists' delays (travel time) and stops (e.g., [6]). However, a few studies have shown that minimizing delays does not necessarily minimize

stops [7]. Considering that FC is highly correlated with stops [3], signal-timing plans that minimize delays do not necessarily lead to optimal reductions in FC, as indicated in several studies (e.g., [7,8]). Therefore, a tradeoff between stops and FC on one side and delays on the other is needed to generate sustainable signal timing.

Balancing between delays and FC in traffic-signal optimization began in the 1970s [9]. However, at that time, traffic-signal timing tools were macroscopic and inaccurate in estimating mobility (e.g., delays) and sustainability (e.g., FC) performance measures. That is because such tools do not accurately capture the individual acceleration traces required to estimate FC and emissions accurately. The past few decades have seen several studies that have used microscopic traffic simulation and FC models to minimize FC and emissions by developing sustainable signal timing [7,10]. Such studies usually utilized heuristic optimization approaches (e.g., a genetic algorithm or "GA") to accommodate the complex nature of microscopic traffic simulations and FC models, which led to near-optimal signal-timing plans.

However, integrating GA optimizers, traffic simulation, and FC and emissions models is a complex process [10]. In addition, such optimization models require processing of every vehicular trajectory, making them very computationally expensive [11]. Furthermore, such integrations are infeasible to implement in the field because of the difficulty of estimating sustainability metrics (FC and emissions) on-site. More importantly, previous studies rarely included the combined effects of multiple factors that impact sustainability metrics (e.g., road grade, percentage of heavy vehicles in the fleet) in the optimization process.

This study attempts to fill the gaps mentioned above by proposing a methodology that optimizes traffic-signal timing (i.e., cycle length, green, and offsets) by minimizing the FC Intersection Control Performance Index (FCIC-PI) presented by Stevanovic et al. [12] as a surrogate measure for FC. This study first investigates the potential fuel savings obtained in using the FCIC-PI in three optimization scenarios under moderate conditions of a real-world test site. Such moderate conditions mean that operating conditions with high impacts on sustainability metrics (e.g., percentage of heavy vehicles in the fleet and road gradient) have a minimal presence; hence, their impact is minor. Furthermore, to illustrate the benefits of the proposed methodology under more diverse scenarios, the authors altered the percentage of heavy vehicles in two other optimization scenarios. We chose to alter the percentage of heavy vehicles because it is one of the most critical operating conditions impacting sustainability metrics. All of the optimization scenarios in this study were performed using the integration of GA optimization (Retime) [13] and a microscopic simulation model (Vissim) [14].

The rest of this study is organized as follows. The Section 2 reviews the relevant literature. An overview of the FCIC-PI derivation is provided in Section 3. The Section 4 presents the proposed methodology. A case study is used to apply the methodology, as explained in the Section 5. Section 6 depicts the evaluations and discusses the results. Finally, conclusions are given in the Section 7.

## 2. Literature Review

A large and growing body of the literature has investigated reducing FC and emissions through retiming signals. Earlier studies have revealed that mobility and sustainability measures cannot be reduced using the same cycle length [9]. Thus, the next batch of relevant studies has focused on converting signal-control formulations into an optimization problem to balance mobility and sustainability measures (e.g., [15,16]). Previous studies used macroscopic [17,18] and microscopic [7,10] sustainability metrics estimates in that endeavor. Studies with microscopic estimates are considered more accurate because they captured instantaneous changes in cruising speed, resulting in accurate FC and emissions estimates.

Various studies reported significantly different FC savings. Such savings could be as low as 1% (e.g., [10]) and reach up to 40.9% (e.g., [8]) when timing plans generated by some studies were compared to those obtained from Synchro and TRANSYT (well-known and widely used signal-timing optimization tools). However, studies that reported low savings

in their sustainability metrics are thought to be more accurate and acceptable because they used high-resolution data, as mentioned earlier. Another logical reason for such reasonable thinking is that more than 15–20% savings seems too high and difficult to accept when the underlying methodology lacks the necessary fidelity and data resolution. That excludes studies where a certain percentage of connected and autonomous vehicles (CAVs) were included in the tested fleet because of the advantages provided through connectivity (e.g., [19]). Hence, one of the contributions of this study is to evaluate the reasonable range of savings in sustainable metrics attained from signal-timing optimization.

Although vehicle type is one of the most impactful factors on sustainability metrics, a few studies [17,19,20] only documented the use of both light-duty vehicles (LDVs) and heavy-duty diesel vehicles (HDDVs). Hence, it is difficult to determine the impact of the presence of heavy vehicles on optimization results. Moreover, many studies (e.g., [21,22]) did not precisely report vehicle-type specification in their case studies. Such a lack of documentation of tested vehicle types adds ambiguity to the results, preventing a meaningful conclusion. Thus, this study contributes to the body of knowledge by evaluating the impact of heavy vehicles in the network on optimization results.

It is crucial to note that none of the previous studies included the impact of road gradient in the optimization, even though several studies documented significant implications for the percentage of grade on sustainability metrics [23,24]. This gap can have significant adverse effects on optimization results. For example, the optimizer might generate a "so-called" optimal signal plan for level-terrain conditions on all links in the optimized corridor or network. In contrast, many links in the optimized network may have a particular slope. In such cases, the generated optimal signal plan is not actually optimal. Thereby, one of the primary advantages of using the proposed surrogate performance measure is the ability to include the impact of grade on the sustainability metrics targeted in the optimization process.

It is worth mentioning that researchers have deployed many combinations of traffic-simulation programs, FC and emissions models, and optimization techniques (e.g., [7,10]). Such a variety of combinations has led to different reported results. Thus, it is almost impossible to meaningfully compare all studies and determine a factual range of savings based on fixed (or a range of fixed) conditions and standards.

Finally, most of the recent signal-optimization techniques and their alternative reinforcement learning approaches (e.g., [25,26]) introduced by the previous studies have not been partially adopted by transportation agencies because of their complexity. Moreover, all recent studies proposed optimization models for mixed vehicular fleets (CAVs and human-driven vehicles), which need a relatively long time to be implemented in the field.

Therefore, there is an urgent need, originated by the accelerating change in the climate, for a methodology based on today's average vehicles, traffic signals, and the optimization tools utilized by transportation agencies. This study attempts to fill the abovementioned gaps by providing a methodology that uses the surrogate measure for FC (FCIC-PI) discussed in the following subsection. Such a measure has the potential to be integrated into the most frequently used signal-timing optimization tools in practice (e.g., Synchro, Vistro, TRANSYT-7F). It is worth noting that none of the previous studies in the literature have made use of the objective functions widely used in the most commonly deployed signal-optimization software programs to focus optimization toward sustainability concerns. Therefore, we emphasize that one of the significant contributions of this study is evaluating the deployment of a practical, objective function in an actual optimization on a realistic coordinated corridor with a relatively large number of intersections. Furthermore, the scientific contributions of this research can be summarized as follows:

- Evaluating a surrogate objective function (the FCIC-PI) that balances mobility and sustainability metrics using high-resolution traffic and sustainability data. Previous studies did not use this objective function for optimization purposes. Additionally, very few signal-timing optimizations were carried out with microscopic simulation models that were calibrated and validated with the same level of detail and rigor.

- That the deployed objective function can account for any operational conditions (e.g., percentage of heavy vehicles) that impact sustainability metrics. This is a non-trivial contribution that has not been seen before, considering the significance of those various operational conditions on sustainability metrics.
- Providing a methodology that reduces the heavy computation loads used by the most notable studies (e.g., [10,11]) in the signal-timing optimization field.
- Assessing the impact of the profound presence of heavy vehicles on side streets on the optimization results. This is a critical contribution because current signal-timing optimization practices aim at providing signal priority to major-road traffic to minimize mobility metrics. Such a practice may ignore the significant sustainability footprint of side streets with high heavy-vehicle traffic.

Specifically, our study optimized the traffic-signal timing of 13 signalized intersections, modeled, calibrated, and validated in the microscopic traffic simulation software Vissim. A GA optimization (Retime), which was proven to generate near-optimal solutions [10,13], was deployed in this study to generate optimal signal plans. Two of the default vehicle types (one LDV and one HDDV) defined in the widely used and accepted Comprehensive Modal Emission Model (CMEM) [27] are used in the relevant scenarios investigated in this study. Road gradients were computed for all movements in the network and included in FCIC-PI computation (discussed later).

### 3. Overview of FCIC-PI

This section provides readers with the necessary background on the objective function optimized in this study. The Performance Index (PI) [28], shown in Equation (1), is one of the most common objective functions used in signal-timing optimization. The critical component of the PI is a weighting factor, "*K*", multiplied by the number of stops to give each stop an equivalent number of seconds in terms of delay; hence, it is also known as the "stop penalty".

Recent research on the *K* factor was carried out to derive a new FC Intersection Control Performance Index (FCIC-PI) based on the original PI [4,12]. This FCIC-PI defines the stop penalty as the number of seconds of delay (idling phase) that consume the same amount of fuel consumed during a stopping event (deceleration–acceleration phases). The interesting aspect of such a definition is that the *K* values for individual links (or movements) include the impacts of different factors (e.g., vehicle type, cruising speed, and road gradient) on FC. Hence, one can use the FCIC-PI as a surrogate measure for actual FC estimates to optimize traffic-signal-timing plans [4,12];

$$PI = \sum_{i=1}^{n} D_i + K \cdot C_i \tag{1}$$

where:

$D_i$—is an average delay in $pcu_s$-hours per hour on the *i*th link (or movement) of the network;
$C_i$—is an average number of $pcu_s$ stops per second on the *i*th link (or movement) of the network;
$n$—is the number of links or movements included in the optimization;
$K$—is the weighting factor, which was defined, for a stopping event, by Stevanovic et al. [12] as

$$K = \frac{(FC_D + FC_A) \cdot T_I}{FC_I} \tag{2}$$

where:

$FC_D$—fuel consumed during the deceleration phase (grams);
$FC_I$—fuel consumed during the idling phase (grams);
$FC_A$—fuel consumed during the acceleration phase (grams);
$T_I$—total idling time (seconds).

One of the significant factors influencing the FC footprint and thus the *K* value at signalized intersections is the percentage of heavy vehicles in a fleet. To show such influence, Figure 1 shows the changes in cruising speed and FC during a complete stop (decelerating from 40 mph, idling for 10 s, and accelerating to 45 mph) for LDVs (Figure 1a) and HDDVs (Figure 1b). It is worth mentioning that Figure 1 was developed based on simulated trajectories from Vissim and FC estimates from the CMEM. In addition, the LDVs and HDDVs used were LDV-category 4 and HDDV-category 7 (described later) in the CMEM, respectively [27]. Two primary observations are apparent in Figure 1. First, the time taken by an HDDV to decelerate from a certain speed to zero and then accelerate to a specific cruising speed is significantly longer than the time needed by an LDV to achieve the same dynamics. Second, the FC rate of an HDDV is higher than that of an LDV for all stop phases (deceleration–idling–acceleration). That is especially seen during the acceleration phase. Thus, the *K* factor of an HDDV is expected to be larger than that of an LDV under identical operating conditions. That means the *K* factor for a particular movement (*i*) changes proportionally to the percentage of HDDVs in the fleet. This study computes the *K* value for a movement as the average of all *K*s calculated for all stops (m) at that movement, as expressed in Equation (3).

$$K_i = \frac{\sum_{j=1}^{m} \frac{(FC_D + FC_A)_j \cdot T_{I_j}}{FC_{I_j}}}{m} \tag{3}$$

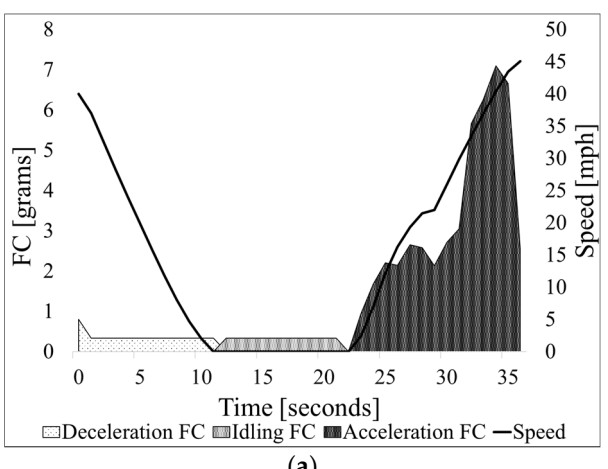

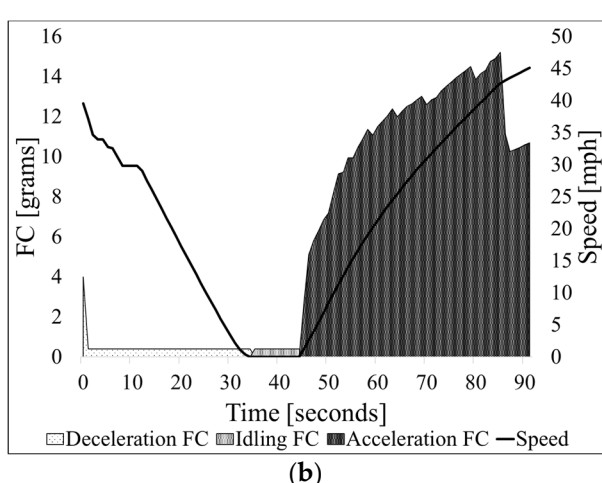

**Figure 1.** Dynamics and kinematics of complete stops made by two types of vehicles: (**a**) LDVs and (**b**) HDDVs.

Based on Equation (3), the relationship between *K*s in various movements would impact how the entire intersection performs in terms of consumed fuel when adjusted for signal timing. A similar statement could be made for the whole network. Therefore, an FCIC-PI for a number of movements, *n*, and a given analysis period (e.g., an hour) for an intersection or a network can be expressed as [4,12]:

$$FCIC - PI = \sum_{i=1}^{n} D_i + \left( \frac{\sum_{j=1}^{m} \frac{(FC_D + FC_A)_j \cdot T_{I_j}}{FC_{I_j}}}{m} \right)_i \cdot S_i \tag{4}$$

where:

*Di*—is the total stop delay at movement *i*;
*Si*—is the total number of stops at movement *i*.

Readers are referred to the relevant papers in the literature to attain more details about Equations (1)–(4).

## 4. Methodology

The methodology started with computing the *K* factor for each movement in the case study investigated in this study (discussed later). The next step was integrating the *K* values into the adopted online optimization tool Retime, which interfaced with a Vissim microscopic simulation. The FCIC-PI was optimized for five optimization scenarios—three moderate conditions and two artificial conditions developed to show the importance of heavy vehicles when developing signal plans to reduce FC and emissions. This study then evaluated the impact of the newly developed signal plans on mobility and sustainability metrics.

### 4.1. Vissim

In this study, the microscopic simulation model PTV Vissim was selected to serve as a stochastic traffic model that suits the stochastic nature of GA optimization. Vissim has been recognized in the traffic community for its friendliness and ability to model the most commonly implemented traffic-signal controllers in the US. Vissim was used in this study at two stages for each of the six performed optimizations as follows: 1. running the base case (unoptimized) model and extracting its second-by-second vehicular trajectories, including timestamp, speed, and acceleration–deceleration traces, to compute the *K* factor for each movement using Equation (3), and 2. providing the mobility measures (e.g., delays) for each possible solution (set of signal plans) provided from the optimization tool to compute the FCIC-PI for that solution, using Equation (4). Finally, the vehicular trajectories of the best-optimized model (set of signal plans) for each scenario were extracted from Vissim and processed in the CMEM (described next) to evaluate the FC and emissions improvements.

### 4.2. CMEM

The CMEM is a microscopic power-demand model that estimates FC and emissions based on various components (e.g., vehicle engine) correlated with vehicles' FC and emissions production. This study used the CMEM to 1. estimate FC measures to compute the *K* factor of all movements using stop profiles' trajectories extracted from the base case (with unoptimized signal plans) simulation models of all five scenarios evaluated in this study and 2. evaluate the improvements in vehicular FC and the emissions of the best set of signal-timing plans. We performed this evaluation by developing a Vissim–Python–CMEM interface (shown in Figure 2) that processed vehicular trajectories from Vissim in the CMEM, providing the total FC and emissions. It is worth noting that previous studies have calibrated and validated sustainability metrics estimates from the CMEM and indicated that the CMEM generates acceptable FC and emission estimates [29,30]. Hence, calibration efforts for the CMEM were not made in this study; instead, two default vehicle types were used. Those vehicle types are (i) car (LDV), normal emitting, three-way catalyst, fuel-injected, >50 k miles, low power/weight and (ii) HDDV 1999–2000, four-stroke, normal emitting. Unlike other studies, we emphasize that our methodology does not estimate FC and emissions for every possible solution provided by Retime during the optimization. Instead, the sustainability improvement is only measured for the optimal set of signal timing (for each scenario) once optimization is completed. This approach is more time-efficient than the most notable previous studies [7,10] because it significantly reduces the evaluation run time of each generation in the optimization.

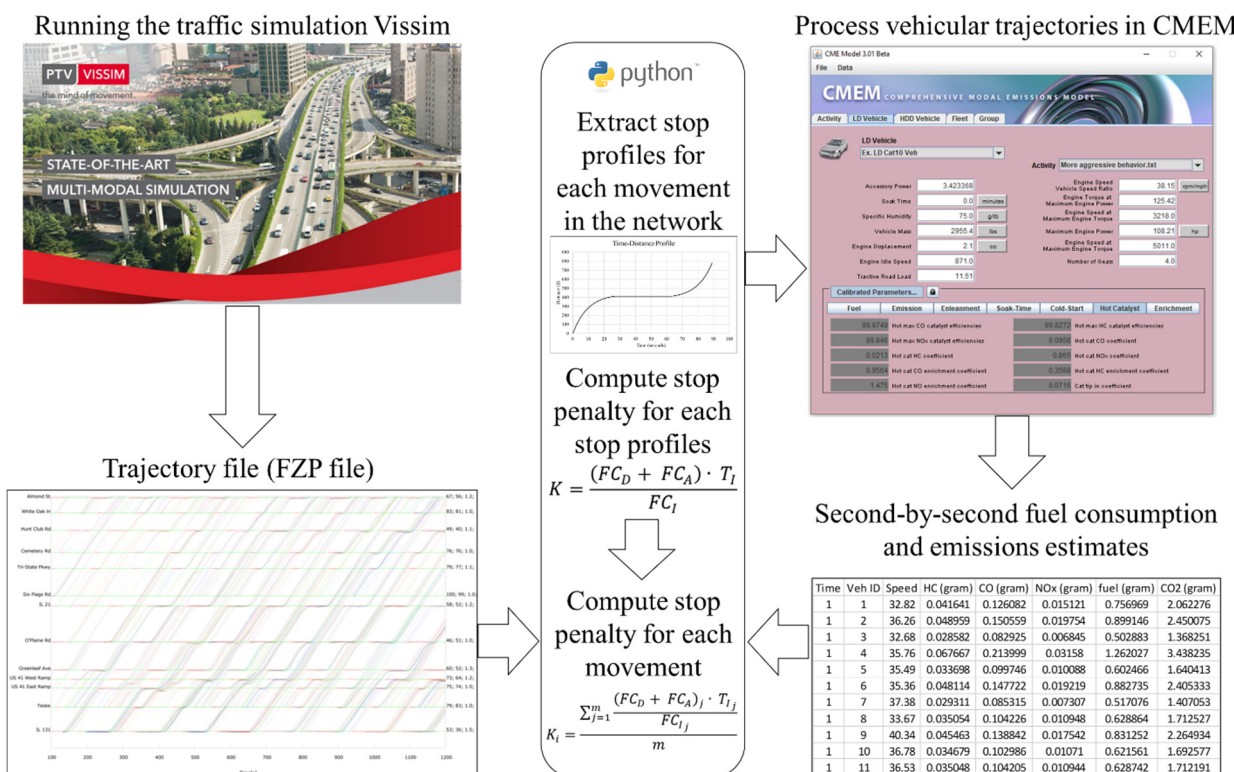

**Figure 2.** The connection between Vissim Version 10, Python Version 3.9, and the CMEM Version 3.0.

### 4.3. Computation of Stop Penalty

The *K* factors were computed by running the base case model for each of the five scenarios before starting the optimization process. As shown in Figure 2, the vehicular trajectories (FZP file) from Vissim were processed in Python to extract the stop profiles' trajectories of all stopped vehicles at each movement. Those trajectories were then processed in the CMEM to estimate second-by-second FC measures during each stop's driving phase (deceleration, idling, acceleration). The FC estimates in each phase were then used to apply Equation (2) and compute the *K* factor for each stop. Once the *K* factor was calculated for all stops individually, Equation (3) was applied to compute *K* for each movement in the network. The final *K* value for each movement included the impacts of four operating factors: vehicle type, fleet distribution, road grade, and cruising speed. Finally, the movements' *K* factors were integrated into the Retime algorithm to calculate the FCIC-PI during the optimization.

### 4.4. Retime Optimization Tool

The Retime optimization tool is an online signal-timing optimization program that utilizes Vissim microscopic simulation to evaluate signal timing proposed by the stochastic nature of the GA. The general structure of Retime is similar to other GA formulations, and it is well-documented elsewhere [13]. Retime uses VISSIM's input and output files to compute the FCIC-PI and provide a new set of signal plans to be evaluated. It can optimize signal-timing parameters (e.g., cycle length, offsets) by finding signal-timing plans that can reduce the FCIC-PI through a predefined number of generations. Retime was extended to accommodate the needs of this study, including functionalities to enable optimization of signal settings to minimize the FCIC-PI for the entire network. Algorithm 1 shows a basic step-description of Retime operations process used in this study.

---

**Algorithm 1:** Description of the genetic algorithm optimization coded in the Retime optimization tool.

---

Step 0: Initializing

    *G*, total number of generations

      *T*, total number of timing plans per generation

      *i*, current number of population

      $i = 0$

    Generation of initial population $p^i$ of timing plans $tp^k$, $\forall k \in [1, \ldots, T]$

- Read initial timing plan $tp^1$ from field Vissim files
- Generate $tp^k$, $\forall k \in [2, \ldots, T]$

Step 1: Evaluating Population

    Evaluation of $tp^k \in p^i$, $\forall k \in [1, \ldots, T]$

- Write $tp^k$ to Vissim files (on the main VM)
- Transfer Vissim files to cloud (to one of the temporary created VMs)
- Simulate and evaluate $tp^k$
- Transfer evaluation results back to the main VM
- Calculate *FCIC-PI$^k$* by applying the following formula for $\forall$intersection, $\forall$movement and $\forall$vehicle class:
  *FCIC-PI$^k$* = $\sum$(stoppedDelayTotal + stopPenalty * stopsTotal) * numOfVehPerClass/*numOfVehiclesOverall*
- Calculate *fitness$^k$* while applying penalty if *externalQueue$^k$* reached the limit

Step 2: Testing Termination Criteria

    *fitness$^b$* = max(*fitness$^1$*, $\ldots$, *fitness$^T$*)

      IF ($i = G$)

          Stop and RETURN $tp^b \in p^i$

    ELSE

          GO TO Step 3

Step 3: Generating New Population

    $i = i + 1$

      Generation of new population $p^i$

- Select a couple of timing plans from $p^{i-1}$ based on their fitnesses (e.g., probabilities to be selected for mating)
- Generate a new couple of timing plans through GA operations (crossover and mutation)
- Continue generating the rest of the $p^i$ by repeating the previous two steps until the required number of timing plans is created

    GO TO Step 1

---

### 4.5. Retime Stochastic Optimizations

Signal-timing optimization process in this research aimed at minimizing the FCIC-PI as the primary objective function for five scenarios as follows:

- **Scenario 1** represented moderate operational conditions, which were the actual conditions from the case-study site (discussed later). This scenario optimized all four signal-timing parameters (cycle length, offsets, splits, and phase sequence). The cycle-length search range was 40–200 s. Scenarios 2 and 3 also represented moderate operational conditions where the differences were in the parameters optimized, the cycle-length search range, and the movements included, as briefly described below.
- **Scenario 2** aimed to fix the phase sequence as it was in the field and narrow the cycle-length search range to 40–160 s. In addition to its importance in revealing the impact of not optimizing the phase sequence and having a narrower cycle-length search range, this scenario was needed because the government agency operating the case-study corridor prefers to have leading left-turns on all signals.
- **Scenario 3** followed the settings of scenario 2 to optimize through-traffic movements only on major and minor streets. This scenario added significant importance to

through-traffic movements, especially on minor streets, compared to the case where left-turn movements on a major street were optimized. That aligns with most transportation agencies' standard practices, prioritizing through-traffic movements to provide good signal coordination.

- **Scenario 4** assumed an increased percentage of heavy vehicles (from 0% to 15%), but only in the vehicular distribution of the side streets. This increase was applied only for four-leg intersections, and only through-traffic movements saw increased heavy-vehicle traffic. It was intended that this scenario would give higher weight to side-street traffic and open side-street greens more to reduce the FC of such heavily loaded truck movements. For example, a standard PI (which consists only of delays and stops) would not pick up such a change, as delays and stops do not contain information about heavy trucks producing extra FC. At the same time, the FCIC-PI is "equipped" to do so through additional "knowledge" embedded in specific *K* factors (of heavy-traffic side-street movements).
- **Scenario 5** also increased the percentage of heavy vehicles in the vehicular distribution, but this time, it was carried out for the westbound direction of the major road. Similarly to the second scenario, the percentage of heavy vehicles was set to 15%. We note that those 15% of heavy vehicles traveled across the entire corridor in the westbound direction, thus representing a heavy traffic route.

The site conditions in scenarios 1–3 put a shallow focus on any operational conditions that could have a significant impact on FC, as documented in other studies [4] (e.g., similar speeds for most movements, level terrain, and a low percentage of heavy vehicles). Consequently, it is not easy in such cases to show the distinctive advantages of using the FCIC-PI as an objective function over other conventional objective functions. For this reason, two hypothetical scenarios (4 and 5) were proposed to include at least one major factor that would give the FCIC-PI a chance to show its potential. Such an impact was found through a higher percentage of heavy vehicles in the fleet distribution. In those hypothetical scenarios, more weight (higher *K* values) was added to the movements with higher percentages of heavy vehicles.

Each optimization run started with the field's initial signal-timing plans. Each optimization scenario was based on the evaluation of the FCIC-PI accumulated during 60 min of simulation time. Each optimization had a minimum of 50 generations, where 20 signal-timing plans were operated through GA procedures for each generation.

## 5. Case Study

Building, Calibrating, and Validating the Vissim Model

The arterial network representing Washington Street (Lake A22), located in Lake County, Chicago, IL, United States, was chosen as a test site to optimize signal timing using the proposed methodology. The arterial network comprises 13 signalized intersections, as shown in Figure 3. Data-collection efforts were required to provide the necessary input to build the microsimulation model and develop various performance measures to calibrate and validate the Vissim model. Such data included traffic volume, traffic-signal timing sheets, turning-movement count, saturation flow rate, travel time, detectors' locations, and intersections' layouts. The data were collected through remote access to Lake County, Division of Transportation (LCDOT) cameras and the LCDOT Automated Traffic Signal Performance Measures (ATSPMs) software platform. The developed microsimulation model resembled field traffic conditions observed on Washington Street during the p.m. peak hour (15:30–16:30).

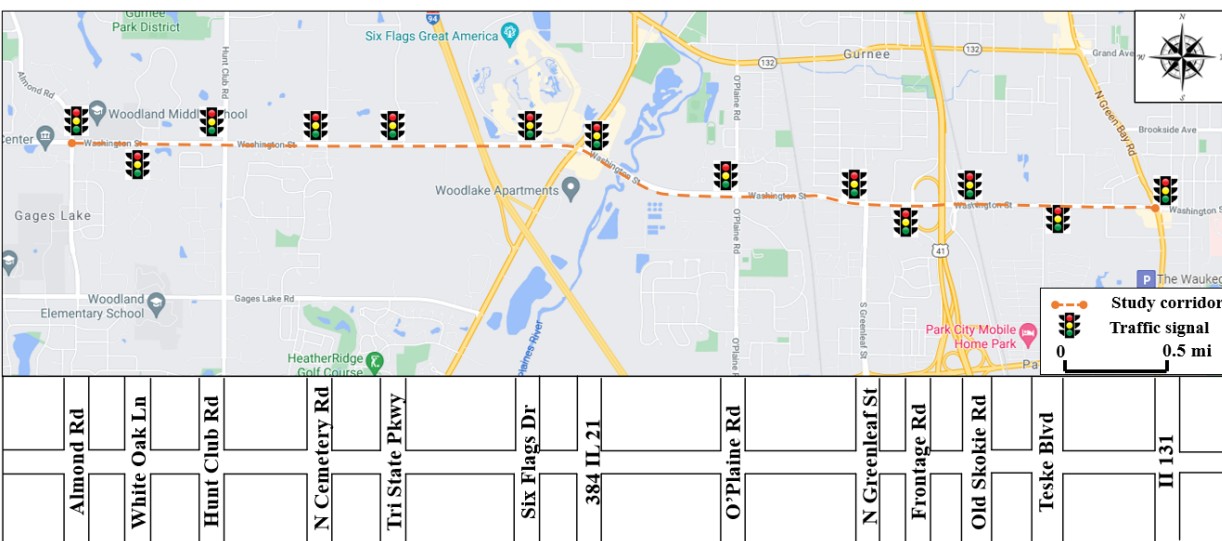

**Figure 3.** The study area of Washington Street.

Traffic volume, saturation flow rate, and travel time were used to calibrate and validate the model. In addition, speed and acceleration–deceleration characteristics were calibrated in this study to reflect realistic driving behavior. That was important because microscopic FC models estimate vehicular FC using acceleration, deceleration, and speed traces. Thus, realistic driving characteristics, such as speed and acceleration patterns, are required to compute an accurate *K* factor.

Vissim defines desired acceleration and deceleration values as distribution functions of the current speed, known as desired acceleration–deceleration functions [14]. Both acceleration–deceleration functions for each vehicle type were defined using three curves representing the minimum, median, and maximum possible acceleration–deceleration values at different speeds. Developing those functions requires an enormous number of field vehicular trajectories covering a wide range of speed. Collecting such trajectories is a costly and time-consuming task. Hence, realistic acceleration–deceleration functions developed in a previous study [31], based on a large-trajectories dataset collected in Michigan, was used for the current testbed. Utilizing those functions from a different location than the testbed's might have resulted in differences between simulated accelerations and decelerations and those happening in the field. However, such differences would be insignificant compared to the differences that would be introduced if Vissim's default acceleration–deceleration functions were used. That is because the default acceleration–deceleration functions in Vissim are based on an older vehicular-trajectories dataset from Europe, as indicated by previous studies [32,33], whereas this testbed is in the US.

Simulated traffic volumes per movement were compared with their counterparts from the field to measure calibration quality. Once calibration was completed, the model was validated against average green times. The model calibration results are shown in Figure 4a, which shows that field traffic volume is highly correlated with modeled volume for the same period.

The correlation between average green times per phase in the field and average green times per phase obtained from the Vissim output file (.lsa file) were utilized to validate the model built in Vissim. The validation results of the average green times are shown in Figure 4b. To further validate the model, link-travel times from the field were compared with the travel times from the model. The results of the travel-time validation are presented in Figure 4c,d for the eastbound (EB) and westbound (WB) directions, respectively. The validation results of the modeled travel times, against their counterparts from the field, showed a strong resemblance between the modeled and the field values. In summary, the calibration and validation results for the modeled period showed that the model

strongly reflects field conditions; hence, it is ready to be used for testing signal-timing optimization scenarios.

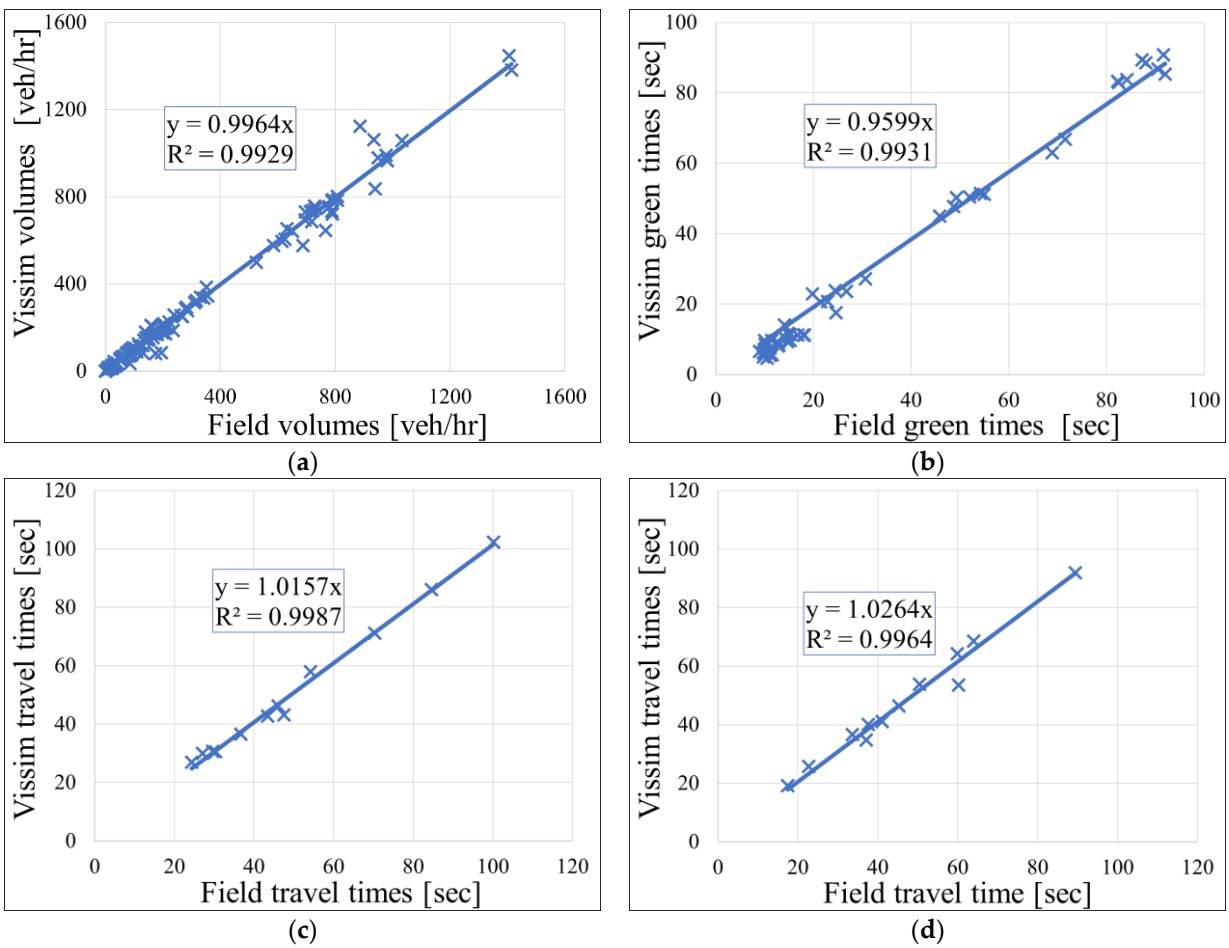

**Figure 4.** Calibration and validation results of the Vissim model: (**a**) traffic volume per movement; (**b**) average green time per phase; (**c**) eastbound link-travel times; and (**d**) westbound link-travel times.

## 6. Results and Discussion

The convergences of the FCIC-PI optimizations for the five scenarios are shown in Figure 5a, demonstrating how the best FCIC-PI varied over the number of generations. The optimization runs for scenarios 1–5 converged after 60, 72, 60, 50, and 50 generations, respectively.

Two observations can be made from Figure 5a. First, the FCIC-PI value at the start of the optimization (number of generations = zero) is significantly different for the five scenarios. Second, scenarios 1–3 have relatively similar convergence patterns, whereas scenarios 4 and 5 resulted in quite different patterns. These observations can be attributed to the high stop-penalty values for the heavy vehicles in scenarios 4 and 5. It is worth noting that the *K* factor for heavy vehicles was ~5 times higher than the *K* factor for light-duty vehicles. It appears that the presence of heavy vehicles, with their high values of *K* factor, can play a significant role in minimization of the FCIC-PI.

The rest of the charts in Figure 5 show how various performance measures changed during various generations of the optimization processes. As expected, as the optimizations progressed, the FCIC-PI was continually reduced, not necessarily in each generation, but whenever a better solution was found. However, improving the FCIC-PI did not constantly improve some of the other performance measures. For example, the conventional Performance Index, delays, and stops (shown in Figure 5) increased for some generations while the FCIC-PI decreased.

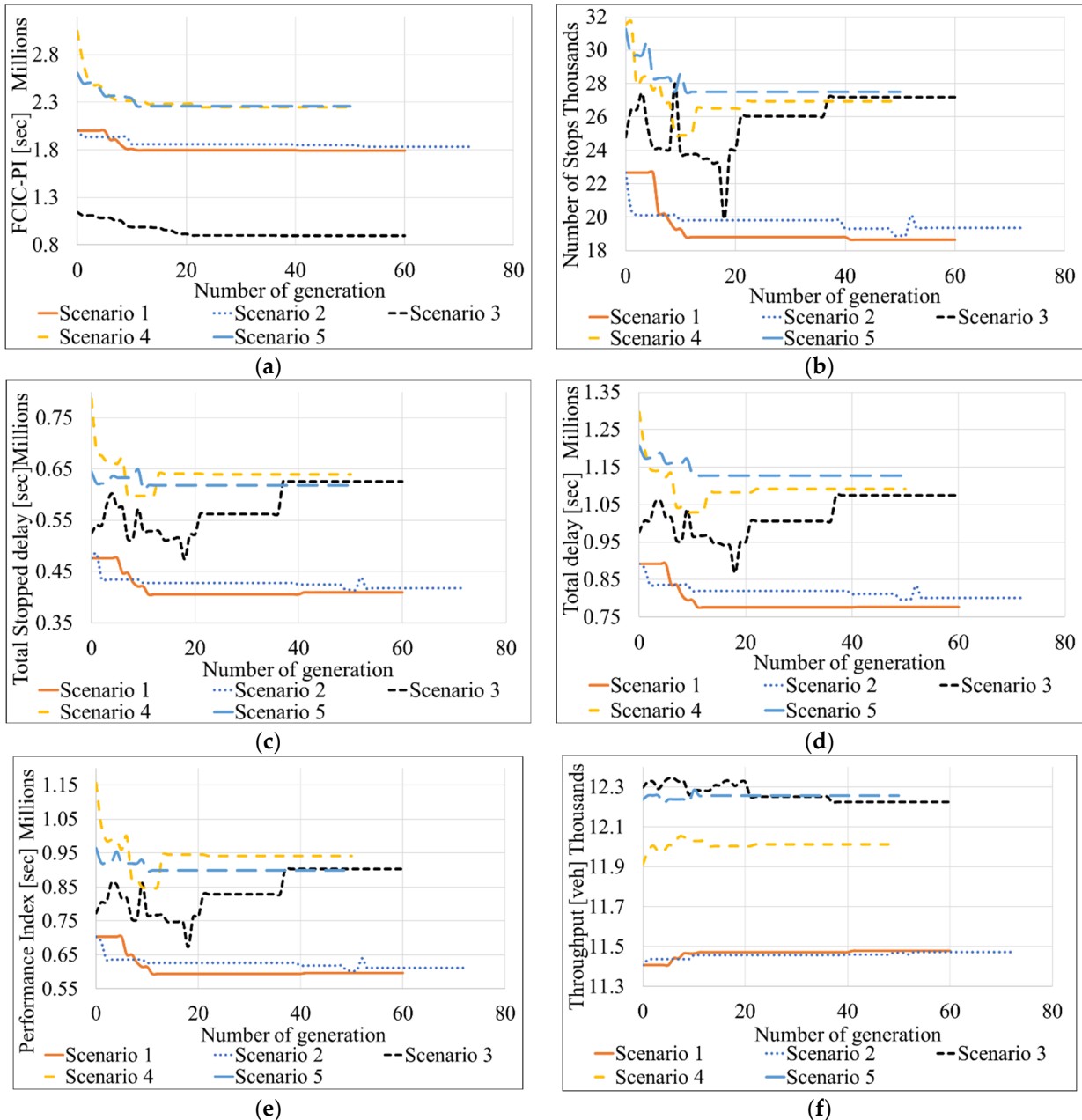

**Figure 5.** Performance-measure (PM) charts—PMs through optimization: (**a**) FCIC-PI; (**b**) number of stops; (**c**) total stop delay; (**d**) total delay; (**e**) traditional Performance Index; and (**f**) throughput.

These inconsistencies in the behaviors of various performance measures can likely be explained with the notion that stops with a high value of stop penalty (*K*) played a significant role in the optimization process. Thus, while the PI considered each stop to be worth the same, the FCIC-PI would give higher weights to those stops whose *K* values were higher. It is also interesting to mention that some optimizations ended up with higher throughputs than those achieved with the initial signal timing, as shown in Figure 5f. This means that the final solution could process more vehicles (which also means more fuel consumed), and it certainly did not keep a significant number of vehicles outside of the network.

One of the most interesting ways to interpret the results is to observe a Pareto chart (shown in Figure 6) illustrating delays and stops as a pair of potentially conflicting performance measures during optimization. The dashed lines represent the paths of the optimal

solutions during the optimization process. Each line starts with a suboptimal combination of stops and delays and keeps moving towards the lower-left corner of the chart, where both stops and delays are minimal. Each dashed line usually ends at one of the points on the red line (Pareto Front), which connects all of the stop–delay combinations where one cannot further improve one performance measure (e.g., stops) without worsening the other (e.g., delays). The fact that the final solution in Figure 6 does not fall on the Pareto Front (red line) is also an indication that the FCIC-PI, as an objective function, takes "something else" into consideration and not only delays and stops; that "something else" is the FC footprints of various stops (e.g., of those experienced by heavy vehicles).

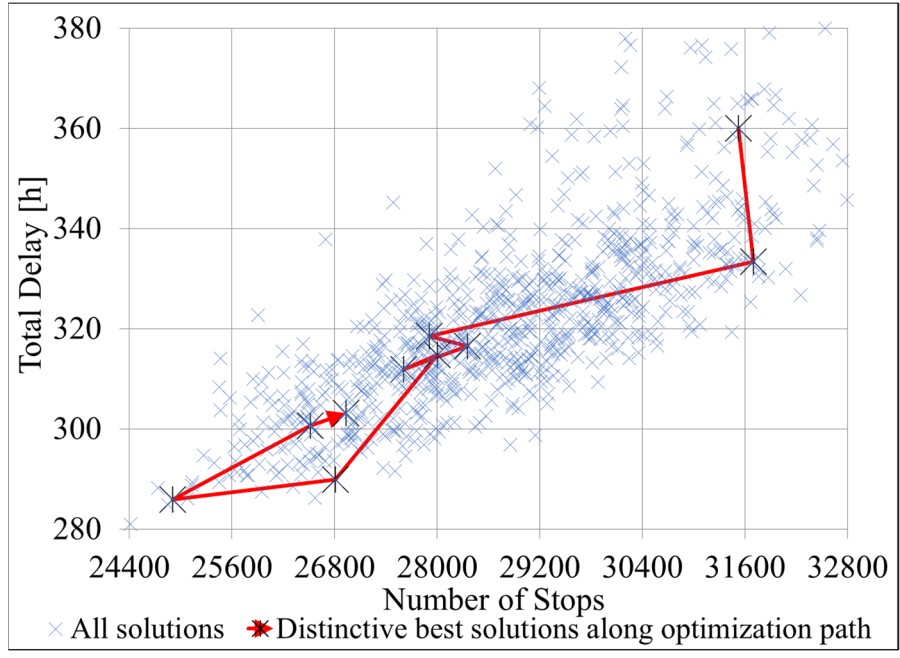

**Figure 6.** Pareto chart—tradeoff between delays and stops for scenario 2.

The current (baseline) signal-timing plans for the optimized intersections ran on a cycle length of 110 s. The west and east through-traffic movements were coordinated to provide progression of those major movements. It is worth mentioning that all left-turn movements led at all intersections and were kept this way during optimization (phase sequence was not included in the optimization) in most scenarios. When it comes to the changes in signal timing that were being made as results of optimization processes, it is clear from Figure 7 that all of the five optimization scenarios led to higher cycle lengths. Specifically, scenarios 1 through 5 show that the cycle length should be increased from 110 s to 118, 115, 122, 119, and 117 s, respectively. All charts in Figure 7 show two values (encircled in red and green) representing cycle lengths before and after optimization. The values encircled with green are for the intersection of Washington Street and IL-131, where the cycle length was 125 s. This cycle length remained constant, as the IL-131 signal was not included in the optimization because it was not coordinated with the rest of the signals. The points encircled in red represent values for all other coordinated signals. Other points in Figure 7 represent other signal-timing parameters (offsets, splits) whose changes cannot be interpreted so easily.

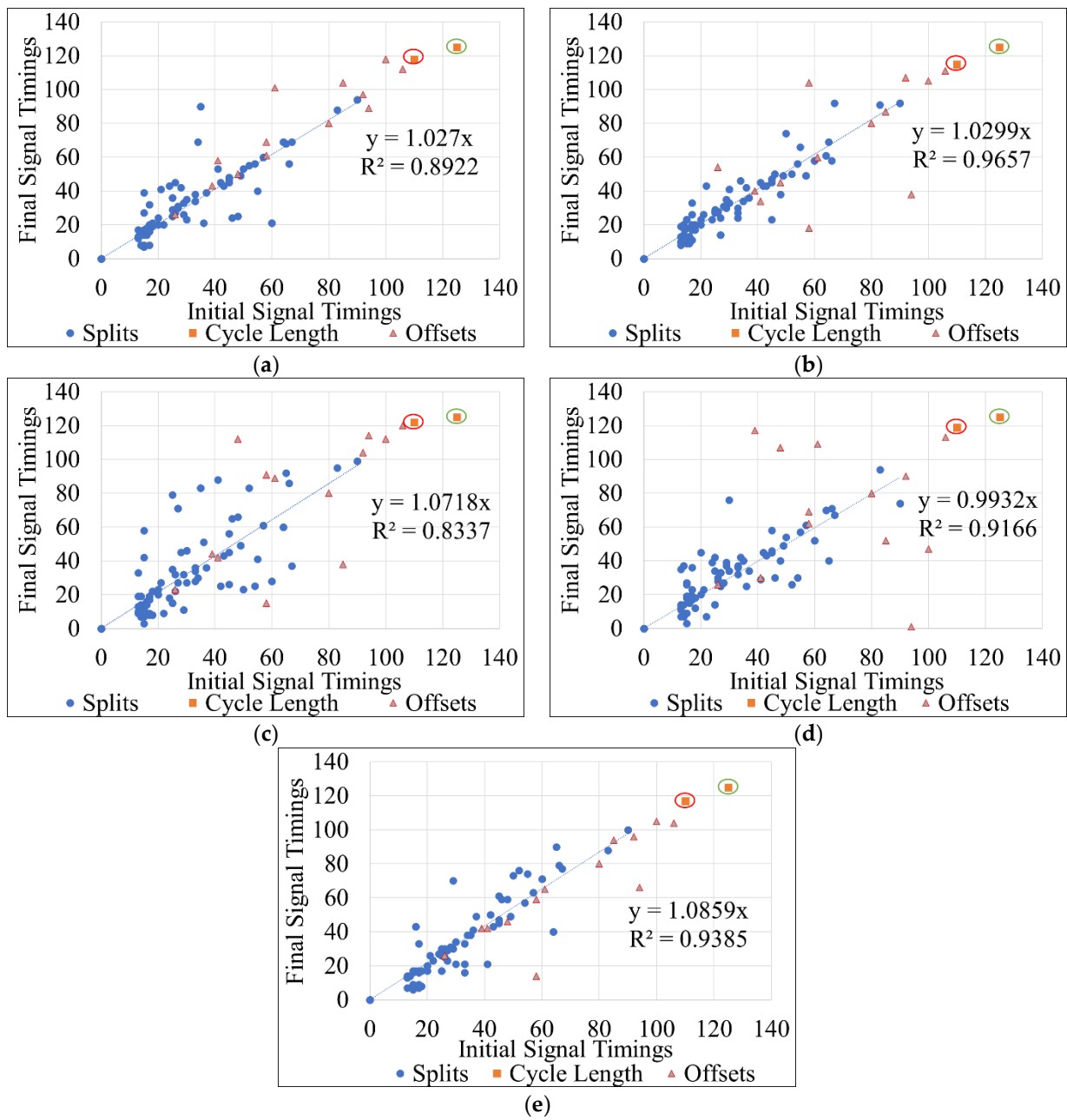

**Figure 7.** Initial vs. final signal-timing parameters: (**a**) scenario 1; (**b**) scenario 2; (**c**) scenario 3; (**d**) scenario 4; and (**e**) scenario 5.

Once the optimizations were completed, the base case and the best model of each tested scenario were used to conduct 50 random-seeded Vissim runs. This step was needed to consider variability in the stochastic nature of traffic in Vissim. Table 1 summarizes the results of 50 simulation runs. While the mobility measures (e.g., delays and number of stops) in Table 1 were obtained directly from Vissim, the FC and emissions were estimated using the CMEM based on individual vehicular trajectories.

**Table 1.** Mobility, fuel consumption, and emissions results from 50-run tests.

| Performance Measure | Statistics | Scenario 1 | | | Scenario 2 | | | Scenario 3 | | | Scenario 4 | | | Scenario 5 | | |
|---|---|---|---|---|---|---|---|---|---|---|---|---|---|---|---|---|
| | | Base Case | Optimized | Mean Difference % | Base Case | Optimized | Mean Difference % | Base Case | Optimized | Mean Difference % | Base Case | Optimized | Mean Difference % | Base Case | Optimized | Mean Difference % |
| Total Delay (hr) | Mean | 245.89 | 233.37 | −5.1 | 245.89 | 242.07 | −1.55 | 245.89 | 241 | −1.99 | 307.7 | 274.3 | −10.85 | 358 | 313.5 | −12.43 |
| | SD | 13.25 | 13.78 | | 13.25 | 13.32 | | 13.25 | 13.24 | | 17.1 | 15.3 | | 17.6 | 17.4 | |
| Stops | Mean | 22,566 | 21,379 | −5.26 | 22,566 | 22,217 | −1.55 | 22,566 | 21,450 | −4.95 | 27,011 | 25,821 | −4.41 | 31,858 | 28,124 | −11.72 |
| | SD | 1941 | 2169 | | 1941 | 2025 | | 1941 | 1952 | | 2358 | 2390 | | 1778 | 2377 | |
| Stopped Total Delay (hr) | Mean | 132.13 | 125.14 | −5.29 | 132.13 | 128.49 | −2.76 | 132.13 | 131.7 | −0.32 | 166.5 | 148.9 | −10.57 | 218.1 | 167.4 | −23.25 |
| | SD | 8.66 | 8.72 | | 8.66 | 8.65 | | 8.66 | 8.69 | | 11.6 | 10.2 | | 13.5 | 11.7 | |
| Throughput (veh) | Mean | 11,384 | 11,407 | 0.2 | 11,384 | 11,409 | 0.22 | 11,384 | 11,396 | 0.11 | 12,187 | 12,185 | 0 | 11,850 | 12,163 | 2.64 |
| | SD | 41 | 45 | | 41 | 44 | | 41 | 43 | | 60 | 55 | | 59 | 61 | |
| Excess FC (g) | Mean | 1,793,948 | 1,604,469 | −10.6 | 1,793,948 | 1,651,540 | −7.9 | 1,793,948 | 1,672,606 | −11.8 | 3,008,700 | 2,590,648 | −13.9 | 2,950,549 | 2,605,509 | −11.7 |
| | SD | 19,821.14 | 17,228.12 | | 19,821.14 | 17,733.55 | | 19,821.14 | 17,959.74 | | 33,242.8 | 27,817.3 | | 32,600.3 | 27,976.87 | |
| Excess HC (g) | Mean | 125,237.2 | 115,021.6 | −8.2 | 125,237.2 | 117,897 | −5.9 | 125,237.2 | 118,602.2 | −9.4 | 151,916 | 137,478 | −9.5 | 166,758 | 158,412 | −5 |
| | SD | 780.35 | 639.94 | | 780.35 | 655.94 | | 780.35 | 659.86 | | 946.59 | 764.87 | | 1039.01 | 881.35 | |
| Excess CO (g) | Mean | 691,580 | 622,150.6 | −10 | 691,580 | 638,202 | −7.7 | 691,580 | 643,881.1 | −11.7 | 892,159 | 778,169 | −12.8 | 942,624 | 884,591 | −6.2 |
| | SD | 7090.06 | 5958.98 | | 7090.06 | 6112.72 | | 7090.06 | 6167.11 | | 9146.39 | 7453.33 | | 9663.76 | 8472.64 | |
| Excess NOx (g) | Mean | 14,258.3 | 13,095.34 | −8.2 | 14,258.3 | 13,348.6 | −6.4 | 14,258.3 | 13,042.13 | −10.9 | 33,597.1 | 28,841.6 | −14.2 | 29,276.5 | 23,946.5 | −18.2 |
| | SD | 155.94 | 139.58 | | 155.94 | 142.27 | | 155.94 | 139 | | 367.45 | 307.41 | | 320.19 | 255.23 | |
| Excess $CO_2$ (g) | Mean | 4,180,735 | 3,723,337 | −10.9 | 4,180,735 | 3,837,664 | −8.2 | 4,180,735 | 3,893,212 | −12.1 | 7,686,980 | 6,574,926 | −14.5 | 7,342,063 | 6,356,867 | −3.4 |
| | SD | 47,702.69 | 41,090.83 | | 47,702.69 | 42,352.55 | | 47,702.69 | 42,965.57 | | 87,709.37 | 72,561.03 | | 83,773.83 | 70,154.53 | |

SD = Standard deviation. Gray shading indicates a 95% significant change. Negative mean difference = improvement in performance. Positive mean difference = degradation in performance.

The improvements in excess FC were between 8 and 12% for scenarios with moderate conditions, whereas the savings reached 12–14% for the scenarios with high percentages of HDDVs. It is worth noting that those savings were computed for the excess FC induced by stops. It is also apparent from Table 1 that emissions do not correlate linearly with each other or with FC. That is because each emission type had a unique savings percentage in various scenarios [34]. The exception is $CO_2$, which seems to have an identical savings percentage to FC. That means that minimizing FC does not necessarily mean minimizing all emission types induced by fuel combustion. Such an observation is essential in seeking an optimal reduction in a specific emission type.

A set of one-tailed Student's *t*-tests, with a confidence level of 99%, was performed to document the statistical significance of the changes in performance measures resulting from the optimized signal plans (compared with the base case signal plans). As shown in Table 1, the test results show that improvements in FC and emissions were statistically significant for most scenarios, with *p*-values < 0.05.

A comparison in the FC savings between scenarios 1 and 2 indicates that optimizing the phase sequence resulted in an extra ~3% FC savings compared to having a fixed lead left turn. Furthermore, narrowing the cycle-length search range did not seem to impact the optimal cycle length, which was far below 160 s for both scenarios.

Optimizing through movements only in scenario 3 resulted in an extra ~4% of savings compared to scenario 2. That could be explained by the fact that through-traffic movements have larger traffic volumes. In addition, through traffic on the major road usually has to accelerate again, if stopped, to a higher speed than those on left turns. That is because the left-turn traffic would proceed to minor streets with lower speeds than the majors. Hence, giving priority to a large traffic volume with a high FC footprint, if stopped, would result in fewer stops, causing less FC.

What is also interesting in Table 1's data is that delays and stops are improved for all scenarios. Even though such improvements were not of the same magnitude as those for FC, they are still statistically significant. Thus, one can conclude that this study's optimal signal plans achieved a good balance between sustainability and mobility measures.

The contributions of this study compared to the relevant studies in the literature can be summarized as follows:

- In most cases, current studies used an isolated hypothetical intersection with a simple phasing design to assess new methodologies for optimizing traffic signals. The outcomes of such methodologies can be considered unreliable because their testbeds did not represent coordination designed in most urban areas. Hence, our study evaluated the proposed optimization methodology by modeling and optimizing a corridor with a realistic scale to what is typically found in the field.

- This study has shown that most emissions do not correlate linearly with FC. The exception was $CO_2$, as discussed earlier in this paper. Such a finding has not been reached by any of the previous relevant optimization studies. It is important to note that the optimization goal may not need to minimize all sustainability metrics simultaneously. Instead, various urban areas or corridors can have different optimization objectives depending on the major detrimental sustainability metric in each area.

- Most of the previous notable studies were time-consuming and required intensive computational loads. Hence, this study evaluated deploying the FCIC-PI, which can be utilized simply by modifying the stop penalties in commonly used optimization programs (e.g., Vistro and Synchro). Such programs are known for their practicality and user-friendly interfaces. It is crucial to note that a few optimization methods were proposed to reduce computational optimization load. Nevertheless, those methods still consume more time and processing loads than what might be practical to implement in the field.

- Many previous articles optimized one or two signal parameters (e.g., cycle length and offset). Such a practice can be partially attributed to operating agencies' policies, which restrict what signal parameters can be modified and by how much. An example is the

LCDOT, the operating agency of the testbed (Washington Street) in this study. The LCDOT does not allow changing of phase sequencing in optimizing signal parameters. For evaluation purposes, however, this study performed a scenario (#1) that optimized all four signal parameters and showed that optimizing the phase sequence resulted in an additional ~3% savings in FC. Such findings might encourage agencies to seriously consider optimizing all signal parameters.

- Although there has been a consensus in the literature on the significant impacts of various operating conditions (e.g., heavy vehicles) on sustainability measures, none of the previous studies has investigated the impacts of such operating conditions on FC-saving results. Hence, one of the significant findings in this study is that including the impact of heavy vehicles on FC in the optimization process results in ~2–4% additional FC savings compared to cases where all vehicles are treated equally in terms of FC footprint. Future research should investigate the impacts of other conditions (e.g., driving behavior, road gradient, etc.).
- Online optimizations have deployed and demonstrated their power in many transportation applications. However, most of the current signal-timing optimization to improve sustainability in the literature cannot be connected to online data sources (e.g., an automated traffic signal performance measure or "ATSPM"). Unlike those studies, our study can simulate large networks with real-time data, as detailed in the following subsection.

### 6.1. Integration of FCIC-PI Methodology in ATSPMs

This subsection explains how the surrogate objective function (the FCIC-PI) evaluated in this paper can be implemented online by being integrated with an ATSPM tool. To compute the FCIC-PI, we need to have stop delay, which is part of an ATSPM system; the number of stops that will be estimated from arrivals on red; and the stop penalty ($K$).

#### 6.1.1. Retrieving Stop Delay

The delay that a typical ATSPM system reports is, most likely, not a stop delay because what is measured in the field is approach delay, which is slightly higher than stop delay, as shown in Figure 8. However, there is a formula from the Highway Capacity Manual (HCM) that can be used to estimate stop delay, stating that the stop delay ($D1$) is approximately equal to the approach delay ($D2$) divided by a factor of 1.3, as shown in Equation (5), where $D2$ is the total approach delay (sec) caused by the traffic signal as reported by ATSPMs. Therefore, the following formula can be used to compute stop delay:

$$D1 = \frac{D3}{1.3} \tag{5}$$

#### 6.1.2. Calculating Stop Penalties

Many connected vehicle data sources (e.g., Wejo and TomTom) provide vehicle trajectories, which can be used to compute the stop penalties for all intersection movements using Equation (3). Although such data sources have a current penetration ratio of less than 10–15%, such a ratio can be representative of the fleet. Thus, their stop penalties can be used in optimization until a higher penetration rate is available. That is especially true when those stop penalties' calculations are paired with a comprehensive knowledge of the field's local conditions. Alternatively, microsimulations can be deployed to extract vehicle trajectories that can readily be used to estimate stop penalties, as demonstrated in this study. Regardless of how stop penalties are computed, they must be entered into the optimization tool before commencing of the optimization process.

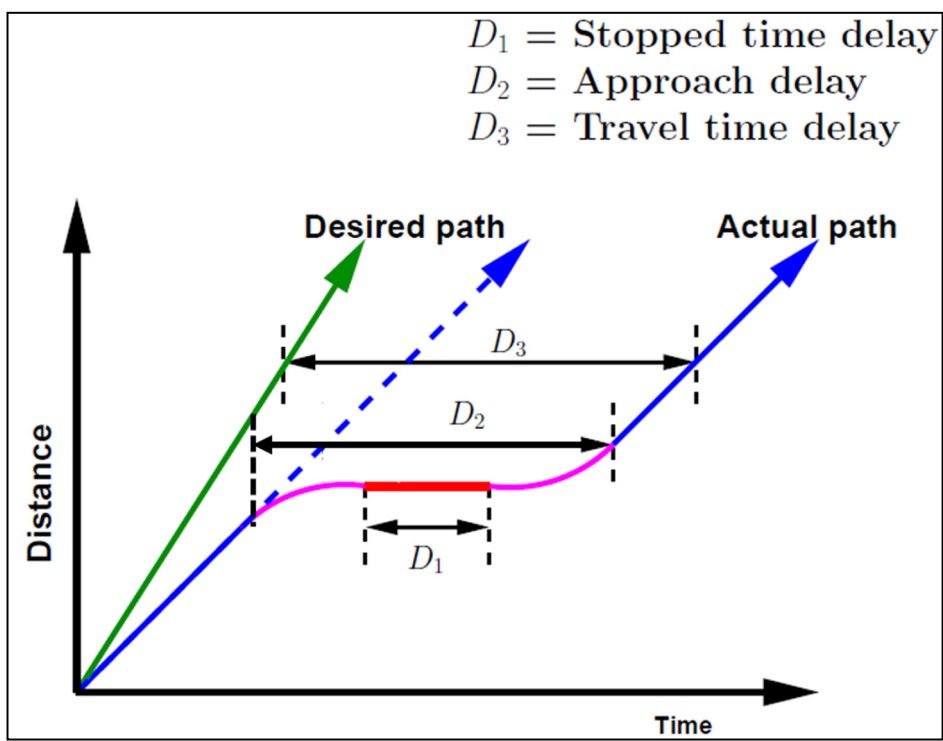

**Figure 8.** Various types of delays caused by a stopping event [35].

6.1.3. Retrieving Number of Stops

For the number of stops, this study proposes using the percentage of arrivals (*AOR*) on red (a performance measure available in a typical ATSPM platform) to compute the number of vehicular stops for relevant movements. *AOR* would need to be multiplied by the traffic volume of a specific movement to calculate the number of stops, as shown in Equation (6), where *V* is traffic volume (veh/hr) and *AOR* is the percentage of arrivals on red (%/100).

$$S = AOR{\cdot}V \tag{6}$$

Using Equations (3), (5), and (6), the total FCIC-PI for the entire system can be computed as follows:

$$FC - PI = \sum_{i=1}^{n} \left[\frac{D2}{1.3}\right]_i + \left[\frac{(FC_A + FC_D)}{\left(\frac{FC_I}{T_I}\right)}\right]_i \cdot [\text{AOR}{\cdot}\text{V}]_i \tag{7}$$

where:

*i*: eligible movement in the network with the estimated approach delay and percentage of arrivals on red;
*n*: total number of eligible movements;
*D*2: approach delay (s);
*FC*: fuel consumption (grams or gallons);
*D*: deceleration (s or hr);
*I*: idling (s or hr);
*A*: acceleration (s or hr);
$T_I$: idling duration (s or hr);
*V*: traffic volume (veh/hr);
*AOR*: percentage of arrivals on red (%/100).

## 7. Conclusions and Future Research

This study aimed to present a new approach to integrating a fuel-consumption surrogate measure, traffic simulation, and signal-timing optimization tools to optimize signal timing under various operating conditions. The proposed approach seeks to achieve a minimal amount of fuel consumption while improving or maintaining efficiency of traffic signals. This study also presented a case study of a network consisting of 13 signalized intersections in Chicago, Illinois, as the test site. For this corridor, five optimization scenarios were completed to determine the fuel-consumption savings obtained using the surrogate objective function, under moderate conditions and with a high percentage of heavy vehicles, for certain intersection movements. Based on the results and the observed findings, the following conclusions were reached:

- The FCIC-PI seems to be a reliable surrogate objective function for fuel consumption when used to optimize traffic-signal timing. This novel PI combines conventional traffic-performance measures (stop delay and number of stops) with a set of factors representing each stop's fuel-consumption weight.
- Using the FCIC-PI saves significant computation time that other approaches must endure because it does not require fuel-consumption estimates from postprocessing vehicular trajectories when optimizing signal timing. Hence, the FCIC-PI can be practical for regular optimizations of signal timing.
- The surrogate objective function used in the optimization resulted in a minimum savings of 8% in fuel consumption. Such savings were significantly increased to 14% (accompanied by significant improvements in the mobility measures) when the impact of a high percentage of heavy vehicles in the fleet was considered in the optimization.

Future research should investigate how well the FCIC-PI captures the impacts of other factors (e.g., road gradients and driving behaviors). Additionally, the savings of the emissions showed that various air pollutants, except $CO_2$, are not linearly correlated with fuel consumption. Thus, further research is needed to investigate the differences between signal optimizations when stop penalties are based on fuel-consumption estimates and when they are based on various emission estimates. A few other limitations of this study are 1. that the impacts of other emission types (e.g., particulate matters) and sustainability objectives (e.g., safety, noise, and pedestrians' and cyclists' exposure to vehicle emissions) were not investigated; 2. that although the CMEM (used in this study) is one of the most reliable fuel-consumption and emission models in the literature, the model is relatively old; and 3. that the acceleration–deceleration functions used in the developed Vissim model were not for the tested location. Therefore, future research should address each of the limitations mentioned above. It is worth noting that addressing those limitations in this study might have increased the findings' accuracy but would not have changed the conclusions.

**Author Contributions:** Conceptualization, S.A. and A.S.; methodology, S.A., A.S. and J.S.; software, J.S. and A.S.; validation, S.A., A.S., J.S. and N.D.; formal analysis, S.A., A.S., J.S. and N.D.; data curation, S.A., A.S., J.S. and N.D.; writing—original draft preparation, S.A.; writing—review and editing, A.S., J.S. and N.D.; visualization, S.A. and A.S. All authors have read and agreed to the published version of the manuscript.

**Funding:** This research was partially supported by funding from the Lake County (IL) Department of Transportation and the research contract on "Traffic Signal Optimization based on Fuel-consumption and Pollutant Emissions". The authors thank the Lake County DOT for their support.

**Institutional Review Board Statement:** Not applicable.

**Informed Consent Statement:** Not applicable.

**Data Availability Statement:** Data used in this research are available upon reasonable request.

**Conflicts of Interest:** The authors declare no conflict of interest.

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
