# Peer review of "Optimizing of Traffic-Signal Timing Based on the FCIC-PI—A Surrogate Measure for Fuel Consumption"

_futuretransp, doi:10.3390/futuretransp3020039_

Round 1
Reviewer 1 Report
1. It is hard to find the contributions of this study. My previous studies have considered the fuel consumption.
2. In section 3, all equations could be found in previous references.
3. In section 4, the simulation scenarios are built by the VISSIM. However, it cannot be defined as the “Methodology”.
4. The current signal timing plans should be explained. In addition, what is the fuel consumption reduction caused by the new signal timing plans?
Author Response
Please, see the responses to your comments in the attached MS Word document.

Reviewer 2 Report
My main concern about the manuscript “Optimizing of Traffic Signal Timings Based on FCIC-PI - a Surrogate Measure for Fuel Consumption” is its contribution to science. Undoubtedly, the research results can improve the transport system's functioning. However, the manuscript lacks a scientific context. It looks more like a presentation of practical solutions than a study of scientific problems. My suggestion is to show the scientific implications in more detail. The aim of the research should be clearly stated. More discussion is needed and comparing the results of the research with similar research known from the literature.
Author Response

(The authors gave the same response as above.)

Reviewer 3 Report
The subject of the article is interesting and the problem is worth exploring.
In the discussion of the results, it would be necessary to refer to other studies in the subject area and point out the differences, advantages and disadvantages of the solutions presented by the authors.
It would be necessary to characterize the authors' barriers and limitations in their experiments.
The purpose of conducting the research and research methods are entirely clear but need clarification:
- it would be reasonable to comment on how the vehicle acceleration variables were calibrated.
- it would be reasonable to comment on whether the baseline (existing condition) scenario was included in the study.
- whether queue lengths and delays were verified for the baseline scenario. If not, what was the reason?
Author Response

(The authors gave the same response as above.)

Round 2
Reviewer 2 Report
Dear Authors,
You have improved the manuscript. I recommend it for publication in a present form.